# Electron Holes in a Regularized Kappa Background

Fernando Haas[1,*], Horst Fichtner[2,*], and Klaus Scherer[2,*]

[1]Physics Institute, Federal University of Rio Grande do Sul, CEP 91501-970, Av. Bento Gonçalves 9500, Porto Alegre, RS, Brazil

[2]Institut für Theoretische Physik, Lehrstuhl IV: Plasma-Astroteilchenphysik, Ruhr-Universität Bochum, D-44780 Bochum, Germany; Research Department, Plasmas with Complex Interactions, Ruhr-Universität Bochum, D-44780 Bochum, Germany

[*]These authors contributed equally to this work.

**Correspondence:** Fernando Haas (fernando.haas@ufrgs.br)

**Abstract.** The pseudopotential method is used to derive electron hole structures in a suprathermal plasma having a regularized $\kappa$ probability distribution function background. The regularized character allows the exploration of small $\kappa$ values beyond the standard suprathermal case, for which $\kappa > 3/2$ is a necessary condition. We have found the nonlinear dispersion relation yielding the amplitude of the electrostatic potential in terms of the remaining parameters, in particular the drift velocity, the wavenumber and the spectral index. Periodic, solitary wave, drifting and non-drifting solutions have been identified. In the linear limit, the dispersion relation yields generalized Langmuir and electron acoustic plasma modes. Standard electron hole structures are regained in the $\kappa \gg 1$ limit.

## 1 Introduction

The phenomenon of so-called electron holes in a plasma has received growing attention in the recent past specially due to recent spacecraft observation of such structures, see, e.g., (Steinvall , 2019a, b) In particular a recent study resolved the phase space density deficit of trapped electrons and proved that the solitary waves with bipolar profiles observed in space plasma are electron holes (Mozer , 2018). General reviews on electron holes can be found e. g. in (Luque , 2005; Eliasson , 2006). For the application to space plasmas a quantitative treatment of electron holes should take into account the presence of a suprathermal, i.e. non-Maxwellian background plasma. This was already pointed out in (Schamel , 2015, 2023) and carried out in (Haas , 2021; Aravindakshan , 2018, 2020; Jenab , 2021). In (Haas , 2021) the Maxwellian description of the trapped (hole) and untrapped (background) electron populations was substituted by one with a so-called standard kappa distribution (SKD).

The SKD is a simple generalization of a Maxwellian that was originally introduced by (Olbert , 1968) to describe non-Maxwellian power-law distributions of suprathermal plasma species, which are frequently observed in the solar wind (Lazar , 2017) and are formed via the interaction of the solar wind particles with the plasma turbulence (e.g., Ma & Summers , 1998; Yoon , 2014; Yoon et al. , 2018) preventing a relaxation to a Maxwellian or bi-Maxwellian. Since then the SKD has been applied successfully to numerous space plasma and laboratory scenarios. Along with these successes also various limitations of the SKD were identified: it exhibits diverging velocity moments, a positive lower limit of allowed kappa parameter values ($\kappa > 3/2$), and a non-extensive entropy (for a recent overview see (Lazar , 2021)). In addition, two types of SKDs were

identified, namely the original one introduced by Olbert (Olbert , 1968) with a prescribed reference speed and a modified one that can be traced to Matsumoto (Matsumoto , 1972) with a temperature equal to that of the associated Maxwellian, and it was demonstrated (Lazar , 2016) that care has to be taken in selecting one of those for a given physical system. The kappa distribution was proposed in (Vasyliunas , 1968); extensive discussion on the different kappa distributions can be found in (Pierrard , 2010; Hau , 2007). Besides these principal limitations of SKDs, there is also an observational one: SKDs do not allow to describe velocity distributions which are harder than $v^{-5}$. However, distributions with harder tails are actually observed, see, for example, (Gloeckler et al. , 2012). At the same time, these measurements also reveal that kappa values near two or below are frequently observed. This can also be seen for solar wind electrons, see, e.g. (Pierrard , 2022). Such low values of kappa imply unphysical features of the SKD as is discussed in (Scherer , 2019). Another example are solar energetic particles, see, e.g., (Oka et al. , 2013). Kappa values as low as 1.63 and two are also obtained for particle distributions in the outer heliosphere (e.g., Heerikhuisen et al. , 2008; Zirnstein et al. , 2017). Finally, SKDs are not consistent with exponential cut-offs of observed power-law distributions of suprathermal proton in the solar wind (Fisk & Gloeckler , 2012).

All of these complications in employing the SKD can be avoided when one uses the *regularized kappa distribution* (RKD) introduced non-relativistically in (Scherer , 2017) and for the relativistic case in (HanThanh , 2022). The RKD exhibits an exponential cut-off of the power at high velocities. Such cut-off is a result of the fact that any acceleration process can only occur on a finite spatial scale and a finite time scale. Consequently, such power law cannot extend to infinity (as in the case of the standard kappa distribution) but must cut-off. The main purpose of the present work is to adopt a regularized version of the SKD and to analyze the consequences. The RKD particularly removes all divergences in the theory and moves the lower limit for the kappa parameter to zero (Scherer , 2019). Both features have consequences for correspondingly described physical systems: in (Yoon , 2014) it was demonstrated that an 'infrared catastrophe' is avoided when using the RKD instead of the SKD and in (Liu , 2020) it was shown that extending the range of kappa values to zero broadens the possible properties of solitary ion acoustic waves in a plasma with RKD electrons. Here the reference value of $\kappa$ is adopted according to Eq. (2) for the SKD.

Since also the first generalization of the analytical treatment of electron holes in an equilibrium plasma to a suprathermal plasma was achieved by employing the SKD (Haas , 2021), the same constraints remain: not all moments of the velocity distribution functions exist and kappa has to be greater than 3/2, thereby potentially preventing the study of a physically interesting regime because harder velocity distributions are observed, see, e.g. (Gloeckler et al. , 2012; Pierrard , 2022) and were associated with observations of various solitary waves (Vasko , 2017). Therefore, the present work revisits the quantitative treatment of electron holes in a suprathermal plasma, where the electron velocity distribution is described with the RKD.

The structure of the paper is as follows: in section II the one-dimensional RKD is defined, in section III various dimensionless variables are introduced, in section IV the method of the pseudopotential is applied and in section V special solutions of the resulting Poisson equation are derived. After an analysis of the corresponding dispersion relation in section VI for homogeneous trapped electrons distributions, the final section VII contains the conclusions of the study.

## 2 One-dimensional regularized $\kappa$ distribution

The starting point (Scherer , 2019; Liu , 2020) is the three-dimensional isotropic regularized $kappa$ distribution (RKD),

$$f_3(\mathbf{u}) = \frac{n_0}{(\pi\kappa\theta^2)^{3/2}\,U\left(\frac{3}{2},\frac{3}{2}-\kappa,\alpha^2\kappa\right)} \left(1+\frac{u^2}{\kappa\theta^2}\right)^{-\kappa-1} \exp\left(-\frac{\alpha^2 u^2}{\theta^2}\right), \tag{1}$$

where $n_0$ is the equilibrium electrons number density, $\kappa > 0$ is the spectral index, $\theta$ is a reference speed, $U$ is a Kummer function of the second kind (or Tricomi function) described in d (Scherer , 2019; Liu , 2020; Abramowitz , 1972), $\mathbf{u}$ is the velocity vector with $u = |\mathbf{u}|$ and $\alpha \geq 0$ is the cutoff parameter.

In the non-regularized limit $\alpha \to 0$ one regains the SKD

$$f_3(\mathbf{u}) = \frac{n_0\,\Gamma(\kappa+1)}{(\pi\kappa\theta^2)^{3/2}\,\Gamma\left(\kappa-\frac{1}{2}\right)} \left(1+\frac{u^2}{\kappa\theta^2}\right)^{-\kappa-1}, \quad \alpha \to 0, \tag{2}$$

where $\Gamma$ is the gamma function, which is positive defined provided $\kappa > 1/2$. For the RKD this constraint is not imposed on $\kappa > 0$.

For the treatment of electrostatic structures it is convenient to define the one-dimensional RKD. For this purpose we use cylindrical coordinates in velocity space and write $u^2 = v^2 + w^2$, where $v$ is the component of the velocity parallel to the electric field and $\mathbf{w}$ contains only the perpendicular velocity components, with $w = |\mathbf{w}|$. In the isotropic case the one-dimensional RKD is

$$\begin{aligned}
f(v) &= 2\pi\int_0^\infty dw\,w\,f_3(\mathbf{u}) \\
&= \frac{2\pi\,n_0\,e^{-\frac{\alpha^2 v^2}{\theta^2}}}{(\pi\kappa\theta^2)^{3/2}\,U\left(\frac{3}{2},\frac{3}{2}-\kappa,\alpha^2\kappa\right)} \int_0^\infty dw\,w\,\left(1+\frac{v^2+w^2}{\kappa\theta^2}\right)^{-\kappa-1} \exp\left(-\frac{\alpha^2 w^2}{\theta^2}\right) \\
&= \frac{n_0\,(\alpha^2\kappa)^\kappa\,e^{\alpha^2\kappa}}{(\pi\kappa\theta^2)^{1/2}\,U\left(\frac{3}{2},\frac{3}{2}-\kappa,\alpha^2\kappa\right)} \Gamma\left[-\kappa,\alpha^2\kappa\left(1+\frac{v^2}{\kappa\theta^2}\right)\right],
\end{aligned} \tag{3}$$

where here $\Gamma$ is the incomplete gamma function of the indicated arguments (Abramowitz , 1972). In other words, $f(v)$ comes from the three-dimensional version after integration over the two perpendicular velocity components.

In the non-regularized limit $\alpha \to 0$ one regains the standard one-dimensional $\kappa$ distribution (Summers , 1991; Podesta , 2005)

$$f(v) = \frac{n_0\,\Gamma(\kappa)}{(\pi\kappa\theta^2)^{1/2}\,\Gamma\left(\kappa-\frac{1}{2}\right)} \left(1+\frac{v^2}{\kappa\theta^2}\right)^{-\kappa}, \quad \alpha \to 0, \tag{4}$$

which is positive definite provided $\kappa > 3/2$.

In the treatment of electrostatic structures, to satisfy Vlasov's equation the distribution function is a function of the constants of motion. In the one-dimensional, time-independent case, the available constants of motion are given by

$$\epsilon = \frac{mv^2}{2} - e\phi, \quad \sigma = \mathrm{sgn}(v), \tag{5}$$

where $\phi = \phi(x)$ is the scalar potential, where $m$ is the electron mass and $-e$ is the electron charge. The sign of the velocity $\sigma = v/|v|$ is an additional constant of motion just in the case of untrapped particles. The energy variable $\epsilon$ can be used to distinguish untrapped ($\epsilon > 0$) and trapped ($\epsilon < 0$) electrons.

In analogy with (Schamel , 1972, 2015, 2023) (where the background is not in the RKD form), presently one starts from Eq. (3) making for the untrapped part the replacement $v \to \sigma \sqrt{2\epsilon/m} + v_0$, where $v_0$ is a drift velocity, defining the distributions of untrapped and trapped electrons according to

$$
f = f(\epsilon,\sigma) = \frac{A\,n_0}{\theta} \quad \left(1 + \frac{k_0^2 \Psi}{2}\right) \left[ H(\epsilon)\,\Gamma\left(-\kappa, \alpha^2\kappa\left(1 + \frac{1}{\kappa\,\theta^2}(\sigma\sqrt{2\epsilon/m} + v_0)^2\right)\right) \right.
$$
$$
\left. + \quad H(-\epsilon)\,\Gamma\left(-\kappa, \alpha^2\kappa\left(1 + \frac{v_0^2}{\kappa\,\theta^2}\right)\right)\left(1 - \frac{\beta\,\epsilon}{m\,\theta^2}\right) \right],
\tag{6}
$$

$$
A \quad = \quad \frac{(\alpha^2\kappa)^\kappa\,e^{\alpha^2\kappa}}{(\pi\kappa)^{1/2}\,U\left(\frac{3}{2}, \frac{3}{2} - \kappa, \alpha^2\kappa\right)},
\tag{7}
$$

where $H(\epsilon)$ is the Heaviside function. The quantities $k_0$ and $\Psi$ are dimensionless variables proportional respectively to the wavenumber of periodic oscillations and to the electrostatic field amplitude, as will be qualified in the following. In addition, $\beta$ is a dimensionless quantity associated to the inverse temperature of the trapped electrons distribution. Unlike singular distributions as in (Schamel , 2015, 2023; Haas , 2021; Schamel , 2018) here the velocity shifted hole distribution is assumed continuous at the separatrix ($\epsilon = 0$) and an analytic function of the energy for both trapped and untrapped electrons. These choices have been made in order to focus on the role of the cutoff parameter $\alpha$ instead of further aspects.

In the non-regularized case, using

$$
(\alpha^2\kappa)^\kappa\Gamma(-\kappa, \alpha^2\kappa\,s) \to \frac{s^{-\kappa}}{\kappa}, \quad \alpha \to 0, \quad \kappa > 0,
\tag{8}
$$

for a generic argument $s$, and

$$
U\left(\frac{3}{2}, \frac{3}{2} - \kappa, \alpha^2\kappa\right) \to \frac{\Gamma(\kappa - 1/2)}{\Gamma(\kappa + 1)}, \quad \alpha \to 0, \quad \kappa > 1/2,
\tag{9}
$$

from Eq. (6) one obtains

$$
f \quad = \quad \frac{n_0\,(1 + k_0^2\Psi/2)}{(\pi\kappa\theta^2)^{1/2}}\,\frac{\Gamma(\kappa)}{\Gamma(\kappa - 1/2)}\left[H(\epsilon)\left(1 + \frac{1}{\kappa\theta^2}(\sigma\sqrt{\frac{2\epsilon}{m}} + v_0)^2\right)^{-\kappa} + \right.
\tag{10}
$$
$$
\left. + \quad H(-\epsilon)\left(1 + \frac{v_0^2}{\kappa\theta^2}\right)^{-\kappa}\left(1 - \frac{\beta\,\epsilon}{m\,\theta^2}\right)\right],
$$

which is the $\kappa$ version of Schamel's distribution that is given in its original form, e.g., in Eq. (4) in (Schamel , 1986) and illustrated in Fig. 1. A slight difference in comparison to the original formulation (Schamel , 1986, 2012) is that here the trapped electrons are described by a linear function of the energy instead of a Maxwellian function.

Finally, the Poisson equation

$$
\frac{\partial^2\phi}{\partial x^2} = \frac{e}{\varepsilon_0}(n - n_0), \quad n = n(\phi) = \int\limits_{-\infty}^{\infty} dv\,f(\epsilon,\sigma)
\tag{11}
$$

is needed, where $\varepsilon_0$ is the vacuum permittivity. An uniform ionic background $n_0$ has been assumed.

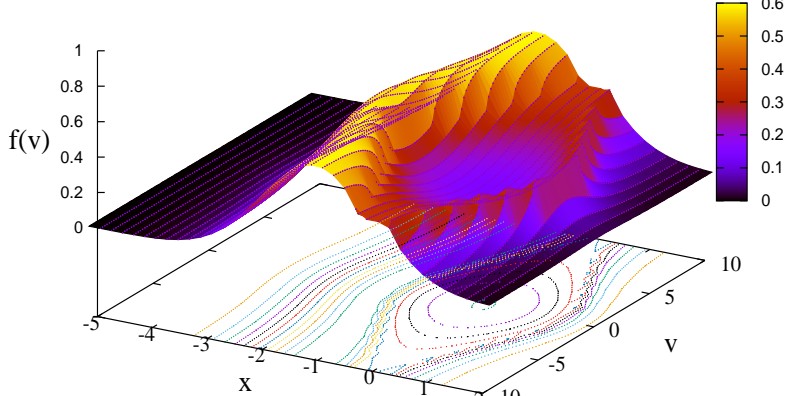

**Figure 1.** An illustration of the Schamel distribution (in arbitrary units and for the sech-potential in Eq.(13) in (Schamel , 1986)) for the values $\beta = -0.9, k_0 = 1.0, \psi = 1.0, \kappa = 0.5, \gamma = 0.1$ of the parameters used for the notation in (Haas , 2021).

## 3 Dimensionless variables

To avoid the use of a large number of parameters, it is convenient to adopt dimensionless variables. For the RKD, it comes the question on which will be the reference speed defining the velocity rescaling. It would be tempting to consider the use of a thermal speed $v_T$ defined in terms of the averaged squared velocity, but it is a cumbersome expression containing Kummer functions,

$$v_T^2 = \frac{<u^2>}{3} = \frac{1}{3} \frac{\int d^3u \, u^2 \, f_3(\mathbf{u})}{\int d^3u \, f_3(\mathbf{u})} = \frac{\kappa\theta^2}{2} \frac{U\left(\frac{5}{2}, \frac{5}{2} - \kappa, \alpha^2\kappa\right)}{U\left(\frac{3}{2}, \frac{3}{2} - \kappa, \alpha^2\kappa\right)}, \tag{12}$$

the factor $1/3$ introduced to comply with the one-dimensional geometry. Therefore, for the sake of simplicity, instead of the thermal speed it is indicated to consider $\theta$ as the reference speed. In this way, the rescaled variables are

$$\tilde{x} = \frac{x}{\lambda}, \quad \tilde{v} = \frac{v}{\theta}, \quad \tilde{v}_0 = \frac{v_0}{\theta}, \quad \tilde{\phi} = \frac{e\phi}{m\,\theta^2}, \quad \tilde{n} = \frac{n}{n_0}, \quad \tilde{f} = \frac{f}{n_0/\theta}, \quad \tilde{\epsilon} = \frac{\epsilon}{m\,\theta^2}, \tag{13}$$

where $\lambda = [\epsilon_0 m\theta^2/(n_0 e^2)]^{1/2}$ is a modified Debye length.

As discussed in (Lazar , 2016) in the non-regularized context, our standard choice of $\theta$ as a $\kappa-$independent parameter better fits a scenario with enhanced tail in velocity space. Alternatively one could choose $v_T$ from Eq. (12) to be $\kappa-$independent, which would be adequate for an enhanced core.

In dimensionless variables omitting for simplicity the tildes, the one-dimensional hole RKD from Eq. (6) is

$$
f(\epsilon,\sigma) = A \quad \left(1+\frac{k_0^2 \Psi}{2}\right)\left[H(\epsilon)\Gamma\left(-\kappa,\alpha^2\kappa\left(1+\frac{1}{\kappa}(\sigma\sqrt{2\epsilon}+v_0)^2\right)\right)\right.
$$

$$
+ \quad \left. H(-\epsilon)\Gamma\left(-\kappa,\alpha^2\kappa\left(1+\frac{v_0^2}{\kappa}\right)\right)(1-\beta\epsilon)\right], \tag{14}
$$

while Poisson's equation (11) is

$$
\frac{\partial^2 \phi}{\partial x^2} = n-1, \quad n=n(\phi)=\int\limits_{-\infty}^{\infty} dv\, f(\epsilon,\sigma), \tag{15}
$$

where $\epsilon = v^2/2 - \phi$ and $\sigma = \mathrm{sgn}(v)$. In the remaining, the purpose is to evaluate the number density in Eq. (15) in terms of $\phi$ and to characterize the possible solutions of the Poisson's equation, specially regarding the behavior according to the parameters $\kappa, \alpha$.

## 4 Pseudopotential method

From Eqs. (14) and (15) one has

$$
\frac{n}{A} = \left(1+\frac{k_0^2 \Psi}{2}\right)\left[\int\limits_{-\infty}^{-\sqrt{2\phi}} dv\, \Gamma\left(-\kappa,\alpha^2\kappa\left(1+\frac{1}{\kappa}(\sqrt{2\epsilon}-v_0)^2\right)\right)+\right.
$$

$$
+ \quad \int\limits_{\sqrt{2\phi}}^{\infty} dv\, \Gamma\left(-\kappa,\alpha^2\kappa\left(1+\frac{1}{\kappa}(\sqrt{2\epsilon}+v_0)^2\right)\right)+ \tag{16}
$$

$$
+ \quad \left. \Gamma\left(-\kappa,\alpha^2\kappa\left(1+\frac{v_0^2}{\kappa}\right)\right)\int\limits_{-\sqrt{2\phi}}^{\sqrt{2\phi}} dv\,(1-\beta\epsilon)\right],
$$

assuming $0 \le \phi \le \Psi$, where $\Psi$ denotes the peak-to-peak amplitude of the electrostatic potential, so that at $\phi = \Psi$ one has $d\phi/dx = 0$.

The integrals in Eq. (16) for the contribution of untrapped particles can be evaluated only in the weakly nonlinear limit. Expanding the integrands in a formal power series on $\sqrt{\phi}$ the result is

$$
n = 1+\frac{k_0^2 \Psi}{2}+a\phi+b\phi\sqrt{\phi}+\mathcal{O}(\phi^2), \tag{17}
$$

keeping the term proportional to $\Psi$ as it has the same order of magnitude of $\phi$ where

$$
a = \frac{2}{\kappa U\left(\frac{3}{2},\frac{3}{2}-\kappa,\alpha^2\kappa\right)}\left[U\left(\frac{1}{2},\frac{1}{2}-\kappa,\alpha^2\kappa\right)+\right.
$$

$$
+ \quad \left.\frac{v_0}{\sqrt{\pi}\kappa}P\int\limits_{-\infty}^{\infty}\frac{ds}{s-v_0}e^{-\alpha^2 s^2}\left(1+\frac{s^2}{\kappa}\right)^{-\kappa-1}\right] \tag{18}
$$

where $P$ stands for the principal value, and

$$b = \frac{4\sqrt{2}}{3}\beta A \Gamma\left(-\kappa, \alpha^2\kappa\left(1+\frac{v_0^2}{\kappa}\right)\right) +$$

$$+ \frac{8\sqrt{2}e^{-\alpha^2 v_0^2}\left[v_0^2 + 2\alpha^2 v_0^4 + \kappa\left(-1+2(1+\alpha^2)v_0^2\right)\right]}{3\kappa^2\sqrt{\pi\kappa}\left(1+v_0^2/\kappa\right)^{\kappa+2}U\left(\frac{3}{2},\frac{3}{2}-\kappa,\alpha^2\kappa\right)}. \tag{19}$$

It is possible to proceed in the same way to determine the average velocity $<v>$ from

$$n\langle v\rangle = \int_{-\infty}^{\infty} dv\, v\, f(\epsilon,\sigma) \tag{20}$$

yielding

$$\langle v\rangle = -v_0\left(1-a\,\phi\right) + \mathcal{O}(\phi^{3/2}) \tag{21}$$

giving a more precise meaning of $-v_0$ which is the global drift velocity only in the limit of zero field amplitude. In addition notice the trapped electrons do not contribute to the average velocity, which comes from the untrapped part only, as found from the detail of the procedure similar to Eq. (16).

Poisson's equation (15) can be rewritten in terms of the pseudopotential $V = V(\phi)$,

$$\frac{d^2\phi}{dx^2} = n - 1 = -\frac{\partial V}{\partial\phi}, \tag{22}$$

where

$$-V = \frac{k_0^2\Psi\phi}{2} + \frac{a\,\phi^2}{2} + \frac{2b\phi^2\sqrt{\phi}}{5} + \mathcal{O}(\phi^3), \tag{23}$$

The case where the solutions are either periodic or solitary waves requires

    1.    $V(\phi) < 0$ in the interval $0 < \phi < \Psi$;

    2.    $V(\Psi) = 0$,

the latter implying

$$k_0^2 + a + \frac{4b\sqrt{\Psi}}{5} = 0, \tag{24}$$

which allows rewriting Eq. (23) as

$$-V = \frac{k_0^2\phi}{2}(\Psi - \phi) + \frac{2b\phi^2}{5}(\sqrt{\phi} - \sqrt{\Psi}), \tag{25}$$

up to $\mathcal{O}(\phi^3)$.

Equation (24) is the nonlinear dispersion relation (NDR) of the problem, providing a relation between phase velocity $v_0$, wavenumber $k_0$ and amplitude proportional to $\Psi$, taking into account the expressions (18) and (19) for $a, b$. On the other hand, Eq. (22) can be integrated yielding

$$\frac{1}{2}\left(\frac{d\phi}{dx}\right)^2 + V(\phi) = 0, \tag{26}$$

where the integration constant was set to zero due to property (I) and since at the potential maximum $\phi = \Psi$ the electric field is zero. Following the usage from (Schamel , 2015, 2023; Haas , 2021; Schamel , 1972, 2018, 1986, 2012), the proposed *Ansatz* has tailored $\Psi$ so that it is the root of $V(\phi)$ in Eq. (25). Otherwise, an irrelevant additive constant would be incorporated in the pseudopotential. The same applies to Eqs. (27) and (31) below.

## 5 Special solutions

### 5.1 Periodic solutions

As discussed in (Schamel , 2015, 2023; Haas , 2021; Schamel , 1972, 2018, 1986, 2012), the expansion of the number density in powers of $\sqrt{\phi}$ starting from an *Ansatz* such as in Eq. (14) can give periodic or localized solutions, according to specific conditions to be identified. For the sake of reference, we collect some of the known analytic solutions, remembering that of course now the coefficients are adapted to the RKD equilibrium. For localized solutions as a by-product one has decaying boundary conditions.

The quadrature of Eq. (26) yields closed form solutions in special cases. In the linear limit, for a small amplitude so that $\sqrt{\Psi} << k_0^2/b$, neglecting the nonlinearity term $\sim b$, one has

$$V = \frac{k_0^2 \phi}{2}(\phi - \Psi). \tag{27}$$

Then from Eq. (26) immediately one has

$$\phi = \frac{\Psi}{2}\left[1 + \cos(k_0\,(x - x_0))\right]. \tag{28}$$

Hence it is verified that $k_0$ indeed corresponds to the wavenumber of linear oscillations with $0 \le \phi \le \Psi$ in this case.

Assuming $k_0 \ne 0$, more insight is provided by the further rescaling

$$\bar{\phi} = \frac{\phi}{\Psi}, \quad \bar{x} = k_0\,x, \quad \bar{V} = \bar{V}(\bar{\phi}) = \frac{V}{k_0^2 \Psi^2}, \quad \bar{b} = \frac{2\,b\sqrt{\Psi}}{5\,k_0^2} \tag{29}$$

reduces Eq. (26) to

$$\frac{1}{2}\left(\frac{d\bar{\phi}}{d\bar{x}}\right)^2 + \bar{V}(\bar{\phi}) = 0, \tag{30}$$

where

$$
\begin{aligned}
-\bar{V}(\bar{\phi}) &= \frac{\bar{\phi}}{2}(1 - \bar{\phi}) + \bar{b}\,\bar{\phi}^2\left(\sqrt{\bar{\phi}} - 1\right) \\
&= \frac{\bar{\phi}}{2}\left(1 - \sqrt{\bar{\phi}}\right)\left(1 + \sqrt{\bar{\phi}} - 2\bar{b}\,\bar{\phi}\right),
\end{aligned}
\tag{31}
$$

containing only one free parameter $\bar{b}$. The condition (II) for periodic or localized solutions amounts to $\bar{V}(\bar{\phi}) < 0$ within the interval $0 < \bar{\phi} < 1$. In view of the factorization in Eq. (31) it is easy to demonstrate the condition is always satisfied for

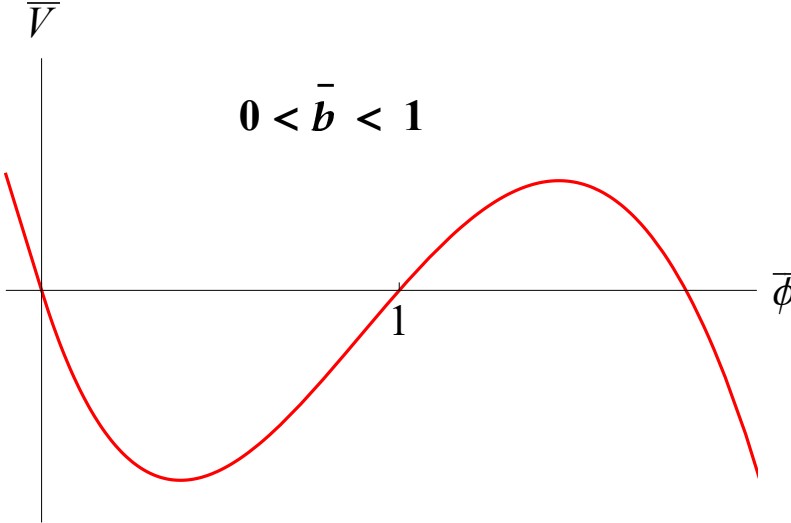

**Figure 2.** Rescaled pseudopotential from Eq. (31) for $0 < \bar{b} < 1$.

$\bar{b} < 1$. The existence of periodic solutions such that $0 \leq \bar{\phi} \leq 1$ for $\bar{b} < 1$ comes from the shape of the rescaled pseudopotential shown in Figs. 2 and 3. The case $\bar{b} > 1$ also has periodic solutions, but with a smaller amplitude as apparent from Fig. 4. The physically meaningful solutions always occur for $\bar{V} < 0$ within the interval $0 < \bar{\phi} < 1$. Notice that with the further rescaling (29) the amplitude of oscillation is set to unity, as shown in the referred figures. The required weakly nonlinear analysis always supposes $\tilde{\phi} \sim \Psi \ll 1$ or, according to Eq. (13), $e\phi/(m\theta^2) \ll 1$, where $\phi$ is the physical scalar potential.

The exact quadrature of Eq. (30) with all terms has been fully discussed in (Schamel , 2012, 2000), where the pseudopotential is formally the same as in Eq. (31) after rescaling. It is given in terms of Jacobi elliptic functions showing a periodic behavior and higher order Fourier harmonics. The present work extends these results for the case of a background RKD, with the adapted coefficients.

It is apparent that the control parameter $\bar{b}$ depending on several variables such as the effective trapped particles inverse temperature $\beta$ determines the qualitative aspects of the oscillatory solutions. Figures 5 and 6 and 7 show in a different style how a smaller (and possibly negative) $\bar{b} < 1$ corresponds to a larger wavenumber, which is exactly $k_0$ only in the linear case.

### 5.2 Localized solution with $\bar{b} = 1, k_0 \neq 0$

The limit case $\bar{b} = 1$ with $k_0 \neq 0$ is special since then $d\bar{V}/d\bar{\phi} = 0$ at $\bar{\phi} = 1$, as shown in Fig. 8, yielding a localized, non-periodic solution. Moreover this case is amenable to the simple quadrature

$$\bar{\phi} = \frac{1}{4}\left[1 - 3\tanh^2\left(\frac{\sqrt{3}}{4}(\bar{x} - \bar{x}_0)\right)\right]^2, \tag{32}$$

see Fig. 9. The corresponding rescaled electric field is shown in Fig. 10. The total electrostatic energy is finite since the integral $(1/2)\int_{-\infty}^{\infty} d\bar{x}(d\bar{\phi}/d\bar{x})^2 = 6\sqrt{3}/35$ converges.

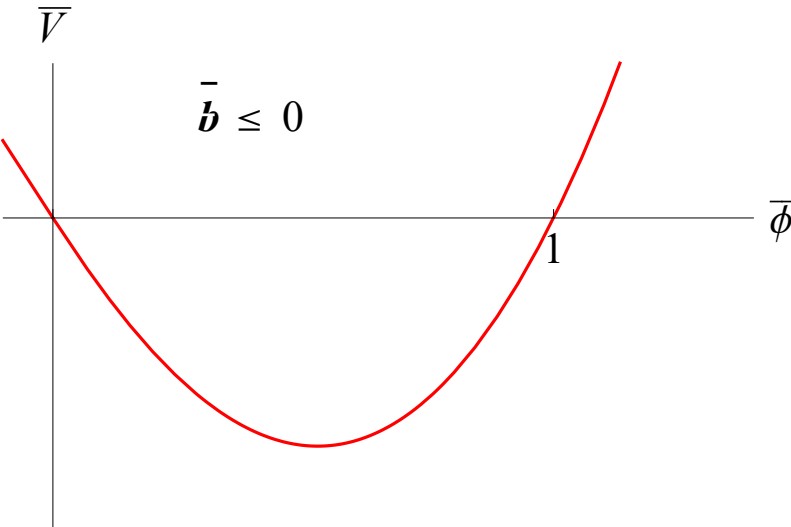

**Figure 3.** Rescaled pseudopotential from Eq. (31) for $\bar{b} \leq 0$.

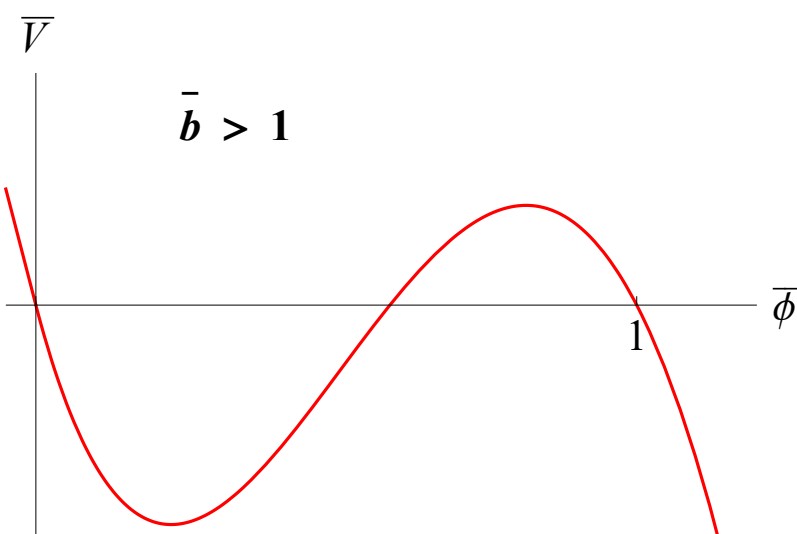

**Figure 4.** Rescaled pseudopotential from Eq. (31) for $\bar{b} > 1$. Periodic solutions exist in a smaller interval $0 \leq \bar{\phi} < 1$.

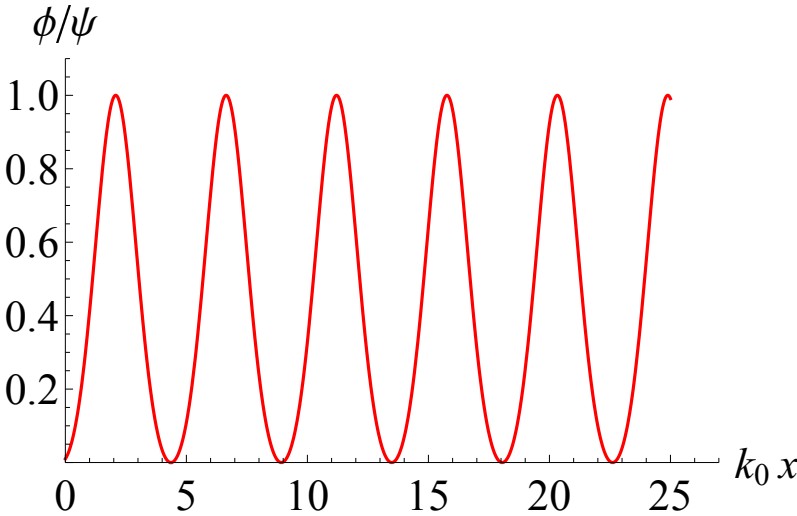

**Figure 5.** Numerical solution of Eq. (30) with $\bar{b} = -2, \bar{\phi}(0) = 10^{-3}$.

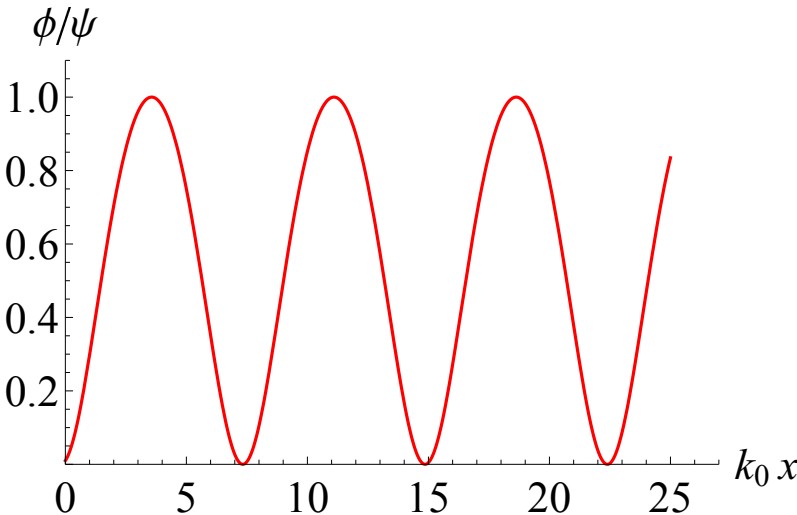

**Figure 6.** Numerical solution of Eq. (30) with $\bar{b} = 0.5, \bar{\phi}(0) = 10^{-3}$.

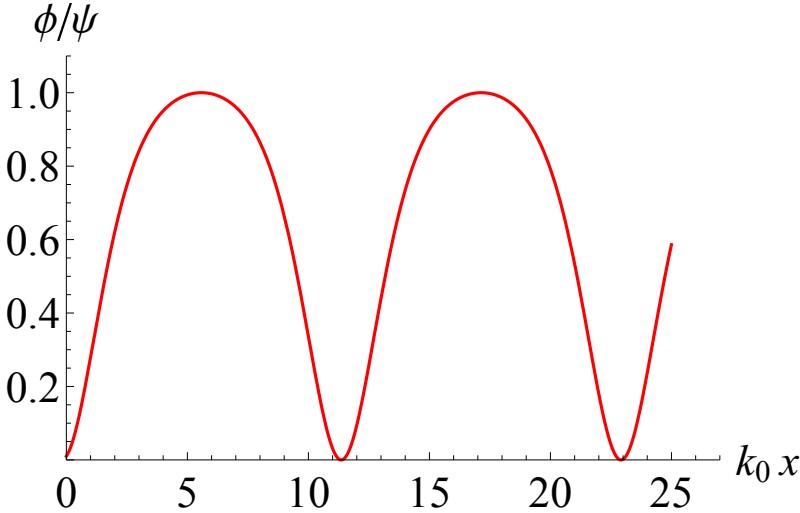

**Figure 7.** Numerical solution of Eq. (30) with $\bar{b} = 0.9, \bar{\phi}(0) = 10^{-3}$.

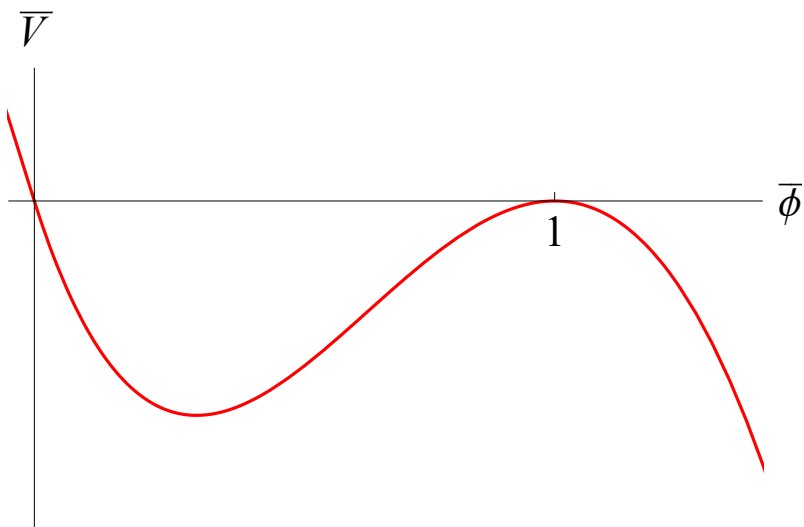

**Figure 8.** Rescaled pseudopotential from Eq. (31) for $\bar{b} = 1$.

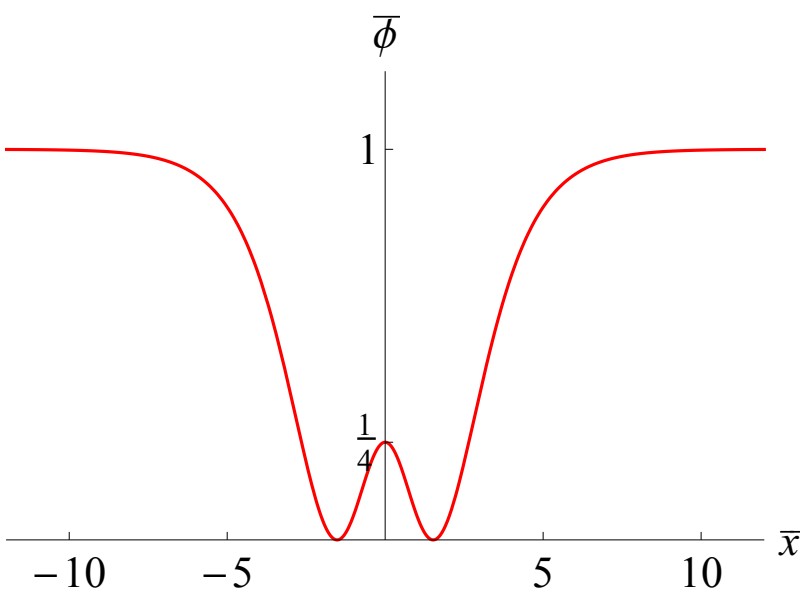

**Figure 9.** Rescaled electrostatic potential from Eq. (32) for $\bar{x}_0 = 0$.

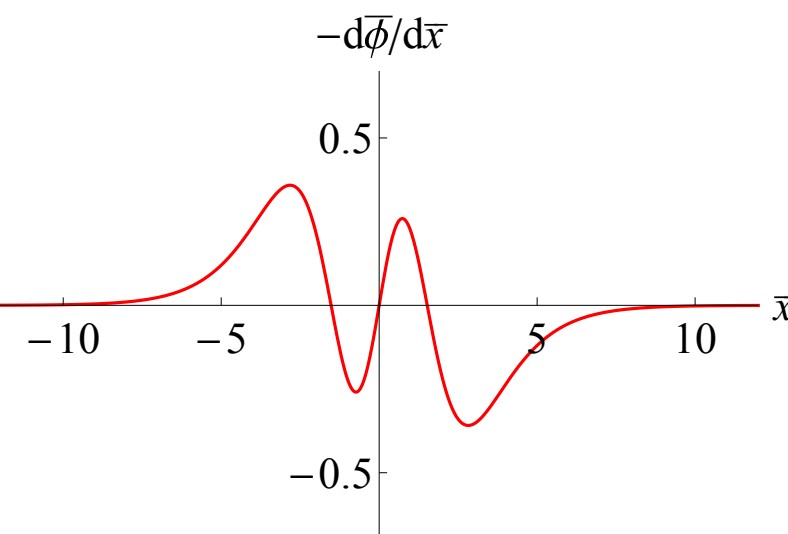

**Figure 10.** Rescaled electric field $-d\bar{\phi}/d\bar{x}$ where $\bar{\phi}$ is given in Eq. (32) for $\bar{x}_0 = 0$.

### 5.3 Solitary waves with $k_0 = 0$.

On the other hand if $k_0 = 0$ one has

$$V = \frac{2 b \phi^2}{5} (\sqrt{\Psi} - \sqrt{\phi}),$$ (33)

yielding the solitary pulse

$$\phi = \Psi \operatorname{sech}^4 \left[ \left( \frac{-b\sqrt{\Psi}}{20} \right)^{1/2} (x - x_0) \right],$$ (34)

which is well defined everywhere provided $b < 0$, which can be attainable e.g. for sufficiently small $\beta, v_0^2$.

## 6 Dispersion relation

The NDR (24) provides several behaviors according to the values in parameter space. For the sake of simplicity it will be considered the case where the trapped particle distribution is homogeneous in phase space, which amounts to the dimensionless quantity $\beta = 0$ in Eq. (10). This is an increasingly better approximation for small enough amplitude so that $e\Psi << m\theta^2$, yielding a relatively smaller trapped area in phase space. Clearly this limit situation does not correspond to "holes", since in this case the trapped particles are not in a depression in phase space as shown e.g. in Fig. 1. However, the analytic simplicity motivates the approach. Furthermore subcases can be identified: drifting, non-drifting; oscillating, non-oscillating, as follows. Our main purpose is to provide an investigation showing a regular behavior for small $\kappa$ values, as long as $\alpha > 0$.

### 6.1 Non-drifting, non-oscillating

If the trapped distribution is homogeneous and non-drifting with respect to the fixed ionic background ($v_0 = 0$), one has from Eq. (24)

$$k_0^2 + \frac{2 U \left( \frac{1}{2}, \frac{1}{2} - \kappa, \alpha^2 \kappa \right)}{\kappa U \left( \frac{3}{2}, \frac{3}{2} - \kappa, \alpha^2 \kappa \right)} - \frac{32 \sqrt{2 \Psi}}{15 \kappa \sqrt{\pi \kappa} U \left( \frac{3}{2}, \frac{3}{2} - \kappa, \alpha^2 \kappa \right)} = 0.$$ (35)

Furthermore in the non-oscillating case $k_0 = 0$ one can solve Eq. (35) as

$$\Psi = \frac{\pi}{2} \kappa \left[ \frac{15}{16} U \left( \frac{1}{2}, \frac{1}{2} - \kappa, \alpha^2 \kappa \right) \right]^2,$$ (36)

which is the amplitude of the solitary wave in terms of the remaining parameters $\kappa, \alpha$ only. Figure 11 shows the resulting amplitude. The regular behavior as $\kappa \to 0$ is apparent. A larger $\alpha$ implies a smaller solitary wave amplitude. In the non-regularized limit $\alpha \to 0$ it is possible to show that from Eq. (36) one has $\Psi \to 1.38$ as $\kappa \to \infty$, which is beyond the weakly nonlinear assumption. From Fig. 11 one also has that the $\alpha = 0$ case only admits small amplitude holes for $\kappa \ll 1$, which is in contradiction with the constraint $\kappa > 3/2$ for the non-regularized equilibrium. It is interesting to note that the weakly nonlinear condition $\Psi \ll 1$ is much better fulfilled for sufficiently high $\alpha$. Hence, such hole structures (with $\beta = 0$, non-drifting and non-oscillating) are more reliable in a RKD background. Note, however, that high $\alpha$ values limit the extent of the power laws.

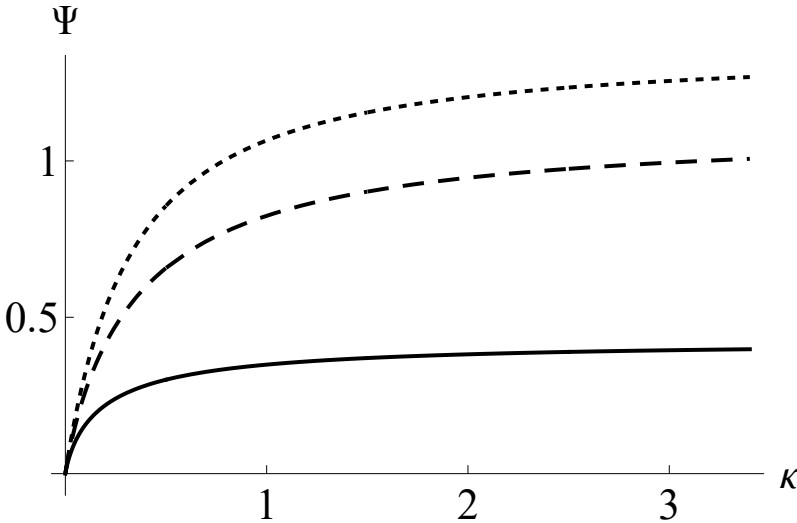

**Figure 11.** Solitary wave amplitude in the homogeneous trapped distribution, non-drifting and non-oscillating case as a function of $\kappa$ and different $\alpha$'s, from Eq. (36). Upper, dotted line: $\alpha = 0.0$; mid, dashed: $\alpha = 0.5$; Lower, solid: $\alpha = 1.5$.

### 6.2 Non-drifting, oscillating

Allowing with $k_0 \neq 0$ for oscillating solutions one also has a regular behavior of the amplitude as $\kappa \ll 1$. In this limit, assuming $\alpha > 0$, it can be shown that Eq. (35) reduces to

$$k_0^2 + \frac{\sqrt{\pi}\,\alpha}{\sqrt{\kappa}} - \frac{32\,\alpha\sqrt{\Psi}}{15\sqrt{2\pi}\,\kappa} = 0, \quad \kappa \ll 1, \quad \alpha > 0 \tag{37}$$

yielding a vanishingly small amplitude as $\kappa \to 0$. Figure 12 shows $\Psi$ from Eq. (35) as a function of $\kappa$, for $\alpha = 1.5$ and different $k_0$ values. It is found that a larger $k_0$ yields a larger amplitude.

### 6.3 Dispersion relation with $v_0 \neq 0$

Allowing for drifting structures so that $v_0 \neq 0$, for simplicity disregarding the nonlinear term $\sim b\sqrt{\Psi}$ and still with homogeneous trapped electrons distribution ($\beta = 0$), one has from Eq. (24),

$$k_0^2 + \frac{2}{\kappa\,U\left(\frac{3}{2}, \frac{3}{2} - \kappa, \alpha^2\kappa\right)}\left[U\left(\frac{1}{2}, \frac{1}{2} - \kappa, \alpha^2\kappa\right) + \right.$$
$$\left. + \frac{v_0}{\sqrt{\pi\kappa}}\,P\int\limits_{-\infty}^{\infty}\frac{ds}{s - v_0}\,e^{-\alpha^2 s^2}\left(1 + \frac{s^2}{\kappa}\right)^{-\kappa-1}\right] = 0. \tag{38}$$

Setting $v_0 = \omega_0/k_0$, Eq. (38) produces similar thumb curves as for holes in a Maxwellian background (Schamel , 1986), now adapted for the RKD. Figure 13 show results for different small $\kappa$ values, in all cases with $\alpha = 0.1$. As usual, one has a high frequency (Langmuir) mode together with a slow electron-acoustic mode (Fried , 1961) now adapted to the RKD background,

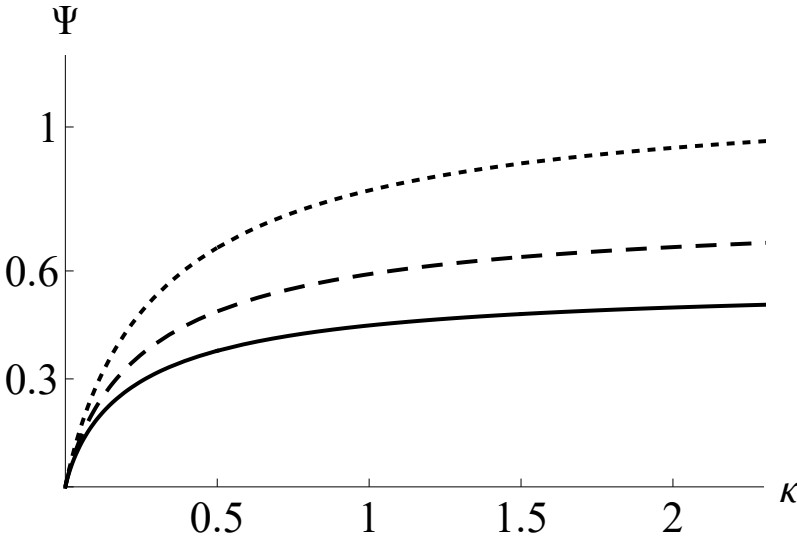

**Figure 12.** Wave amplitude in the homogeneous trapped distribution, non-drifting and oscillating case as a function of $\kappa$ and different wavenumbers, for $\alpha = 1.5$, from Eq. (35). Lower, solid: $k_0 = 1.0$; mid, dashed: $k_0 = 1.5$; upper, dotted line: $k_0 = 2.0$.

where both modes coalesce in a certain point according to the parameters. As seen, the behavior is regular even for small $\kappa$ values. At the extremal $k$ value where both modes coalesce, apparently the group velocity is infinite. As discussed in (Schamel , 2013; Valentini , 2012), at this point taking into account the nonlinear trapping the phase velocity of the hole should replace the diverging linear group velocity.

## 7 Conclusions

In the present paper electron holes have been discussed, for the first time in a suprathermal plasma described with a regularized kappa distribution. Unlike (Haas , 2021), for simplicity, here the background distribution function has no singular features. It was verified that the regularization of the standard kappa distribution avoids all divergent features the solutions for $\kappa \leq 3/2$, i.e. the analysis could be extended to all positive kappa values. This allows one to study plasma backgrounds that are described with velocity power-laws harder than $v^{-5}$ and those that exhibit an exponential cut-off, which are both observed in the solar

wind. Note also, that even for kappa values below two, which can technically be handled with an SKD, unphysical features related to a non-negligible contribution of particles at high velocities are unavoidable (Scherer , 2019). Their removal also requires the use of an RKD.

In terms of the hole distribution function for trapped and untrapped electrons, the number density has been evaluated yielding the pseudopotential in the weakly nonlinear limit. As a consequence, the most prominent solutions of the resulting Poisson

equation have been found. Drifting, non-drifting, oscillating and non-oscillating solutions have been discussed. The linear dispersion relation has been also analyzed, yielding a $\kappa$-dependent plasma mode diagram revealing the existence of a high

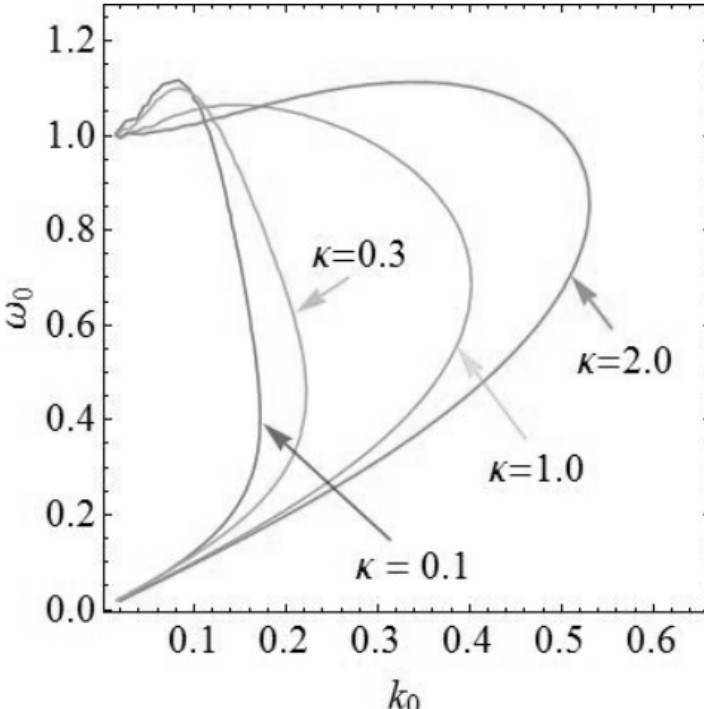

**Figure 13.** Dispersion relation (38) with $v_0 = \omega_0/k_0$ for $\alpha = 0.1$ and $\kappa = 0.1, 0.3, 1.0, 2.0$, as indicated.

frequency Langmuir mode and a low frequency electron acoustic mode (Fig. 13). Unlike for the case of a pure power-law, i.e. for an SKD background, all findings based on power-laws with an exponential cut-off, i.e. based on an RKD background, remain regular even for very small $\kappa$ values. The results are, therefore, relevant especially for those plasmas in a suprathermal
equilibrium state with spectral index $\kappa < 3/2$, for which the SKD is not appropriate, but which are observed in space plasmas (Gloeckler et al. , 2012).

*Data availability.* The data that support the findings of this study are available from the corresponding author upon reasonable request.

*Author contributions.* The authors contributed equally to this work, regarding original idea, basic theory and applications.

*Competing interests.* The authors report no conflict of interest.

*Acknowledgements.* FH acknowledges the support by Conselho Nacional de Desenvolvimento Científico e Tecnológico (CNPq) and the Alexander von Humboldt Foundation for a renewed research stay fellowship.

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
