# Peer review of "Electron Holes in a Regularized Kappa Background"

_Nonlinear Processes in Geophysics, 2023_

## Author Response (AR1)

**Federal University of Rio Grande do Sul**    **Brazil**

Physics Institute    91501-970 Porto Alegre, RS
Plasma Physics Group    Phone: +55 51 3308 6507
Dr. Fernando Haas    Fax: +55 51 3308 7286
    E–mail: fernando.haas@ufrgs.br

June 13, 2023

Prof. Giovanni Lapenta
Editor
Nonlinear Processes in Geophysics

Dear Prof. Lapenta:

We would like to thank you very much for the editorial handling of our manuscript NPG-2023-6, entitled "Electron Holes in a Regularized Kappa Background" by F. Haas, H. Fichtner and K. Scherer.

We are grateful to the Referees for reading our work carefully and for offering constructive comments and suggestions to improve the discussion and ideas presented in the manuscript.

In what follows, we exhibit our point-by-point responses to the Referees' comments.

**Report of the Referee # 1**

- RC1: 'Comment on npg-2023-6', Anonymous Referee 1, 05 Apr 2023

  The authors present a theoretical study of electron phase space structures in the form of periodic and non-periodic 'electron holes' including a trapped portion of electrons in a wave potential. In particular the authors focus on distribution functions with heavy power-law velocity tails, described by Kappa distributions. To avoid the divergence of certain moments they include cut-off function via a Gaussian multiplying the distribution function. The study has relevance to space plasma where localized bipolar electric fields are frequently observed by satellites. I find the paper well written, mathematically correct, and worthy of publication.
  A few minor suggestions are:
  1) Can the authors discuss the physical processes by which the kind of distribution functions they are considering are formed in space plasmas, i.e. why not always Maxwellian (or bi-Maxwellian) distributions?

  **Reply:** these distributions are frequently observed in the solar wind (e.g., Lazar et al. 2017) and are formed via the interaction of the solar wind particles with the plasma turbulence (e.g., Ma & Summers 1998, Yoon 2014, Yoon et al. 2018) preventing a relaxation to a Maxwellian or bi-Maxwellian. We have added corresponding information in the manuscript along with these references.

2) A few review articles on similar topics that the authors could cite for completeness include:
A. Luque, H. Schamel, Electrostatic trapping as a key to the dynamics of plasmas, fluids and other collective systems, Physics Reports, Volume 415, Issues 5-6, 2005, Pages 261-359
B. Eliasson, P. K. Shukla, Formation and dynamics of coherent structures involving phase-space vortices in plasmas, Physics Reports, Volume 422, Issue 6, 2006, Pages 225-290

**Reply:** we have included these references, cited on page 1 in the revised version.

**Report of the Referee # 2**

- RC2: 'Comment on npg-2023-6', Anonymous Referee 2, 24 Apr 2023

In general, I found the manuscript to be well-written and organized, with a clear presentation of the research question, methods, results, and discussion. The literature review is comprehensive, and the analysis appears to be appropriate for addressing the research question.
However, there are a few areas that require further clarification and improvement. My comments/suggestions are below.
1. It will be interesting to the readers, if you explain about the physical scenarios that standard kappa distribution function is unequipped to explain, and we have to use RKD in that case.

**Reply:** we have added to the introduction a few sentences regarding the fact that the SKD cannot be used to describe all observed velocity distributions: power laws harder than $v^{-5}$ and power-laws with exponential cut-offs cannot be modelled with SKDs. However, distributions with such hard tails and power-laws with exponential cut-offs are actually observed as shown in the newly added references.

2. It would better if you can add a sentence and a reference regarding the Kummer function.

**Reply:** we followed the Referee suggestion and included as a reference the handbook by Abramowitz and Stegun, cited on a sentence on page 2.

3. Whether the one-dimensional form of regularized kappa function is does not exists or not? If not, it will nice if you can provide a little more insights towards the derivation. If yes, please cite the reference.

**Reply:** to the best of our knowledge the one-dimensional form of the RKD has not been considered before. Therefore, we followed the recommendation by the Referee and included one more line in Eq. (3), together with a reference to the handbook by Abramowitz and Stegun.

4. The authors mention that the results described in the manuscript are relevant especially for plasmas having a suprathermal equilibrium with small spectral $\kappa$ index, for which the SKD is not appropriate, as frequently happens in space plasmas. But especially in the Pierrard, 2002 the data point are only for $\kappa > 3/2$ (fig2,). It will be nice validation towards to the importance of this work, if you include actual observation of events with $\kappa < 3/2$, where this theoretical model is necessary.

**Reply:** we have added (see our reply to point 1. above) the reference to Gloeckler et al. (2012) where the observation of such hard-tailed distributions are reported (see, particularly, Fig. 1(c) therein).

5. A more detailed explanation is required in the discussion by comparing with SKD.

**Reply:** we have re-written the final section in order to emphasize the improvements made by using an RKD instead of an SKD.

We thank the Referees for reading our manuscript carefully and for pointing out valuable comments and suggestions that have helped us to improve the manuscript and clarify some points. Changes are highlighted in blue in the revised version. We hope that this revised version is now suitable for publication in Nonlinear Processes in Geophysics.

Yours sincerely,

F. Haas (on behalf of the authors)